# An adaptive, asymptomatic SARS-CoV-2 workforce screening program providing real-time, actionable monitoring of the COVID-19 pandemic

Diego R. Hijano [1]⊕*, James M. Hoffman[2,3]⊕, Li Tang[4], Stacey L. Schultz-Cherry[1], Paul G. Thomas[5], Hana Hakim[1], Richard J. Webby[1], Randall T. Hayden[6‡], Aditya H. Gaur[1‡], St. Jude COVID-19 Employee Screening Program Team¶

1 Departments of Infectious Diseases, St. Jude Children's Research Hospital, Memphis, Tennessee, United States of America, 2 Departments of Pharmaceutical Sciences, St. Jude Children's Research Hospital, Memphis, Tennessee, United States of America, 3 The Office of Quality and Patient Care, St. Jude Children's Research Hospital, Memphis, Tennessee, United States of America, 4 Departments of Biostatistics, St. Jude Children's Research Hospital, Memphis, Tennessee, United States of America, 5 Departments of Immunology, St. Jude Children's Research Hospital, Memphis, Tennessee, United States of America, 6 Departments of Pathology, St. Jude Children's Research Hospital, Memphis, Tennessee, United States of America

⊕ These authors contributed equally to this work.
‡ RTH and AHG also contributed equally to this work.
¶ Membership of the St. Jude COVID-19 Employee Screening Program Team is provided in the Acknowledgments.
* diego.hijano@stjude.org

**Data Availability Statement:** All relevant data are within the manuscript.

## Abstract

COVID-19 remains a challenge worldwide, and testing of asymptomatic individuals remains critical to pandemic control measures. Starting March 2020, a total of 7497 hospital employees were tested at least weekly for SARS CoV-2; the cumulative incidence of asymptomatic infections was 5.64%. Consistently over a 14-month period half of COVID-19 infections (414 of 820, total) were detected through the asymptomatic screening program, a third of whom never developed any symptoms during follow-up. Prompt detection and isolation of these cases substantially reduced the risk of potential workplace and outside of workplace transmission. COVID-19 vaccinations of the workforce were initiated in December 2020. Twenty-one individuals tested positive after being fully vaccinated (3.9 per 1000 vaccinated). Most (61.9%) remained asymptomatic and in majority (75%) the virus could not be sequenced due to low template RNA levels in swab samples. Further routine testing of vaccinated asymptomatic employees was stopped and will be redeployed if needed; routine testing for those not vaccinated continues. Asymptomatic SARS-CoV-2 testing, as a part of enhanced screening, monitors local dynamics of the COVID-19 pandemic and can provide valuable data to assess the ongoing impact of COVID-19 vaccination and SARS-CoV-2 variants, inform risk mitigation, and guide adaptive, operational planning including titration of screening strategies over time, based on infection risk modifiers such as vaccination.

**Funding:** SSC, PGT, and RJW are partially supported by the National Institutes of Health, under CEIRS contract no. HHSN272201400006C, SSC under CIVIC contract no.75N93019C00052 and PGT under R01 AI121832. The described work was partially supported by the American Lebanese Syrian Associated Charities (ALSAC). There was no additional external funding received for this study. The funders had no role in study design, data collection and analysis, decision to publish, or preparation of the manuscript.

**Competing interests:** PGT is on the Scientific Advisory Board for Cytoagents Inc. RTH has served on advisory boards for Roche Molecular and Quidel Corporation. All other authors have no conflicts of interest. This does not alter our adherence to PLOS ONE policies on sharing data and materials.

## Introduction

The COVID-19 pandemic continues. SARS-CoV-2 variants have emerged and affected morbidity, mortality and overall efficacy of the first-generation COVID-19 vaccines [1, 2]. Pandemic fatigue combined with a sense of security amongst growing numbers of those vaccinated has led to relaxation of mitigation strategies in some countries and increases in cases. This has again led to increased localized utilization and subsequent burden on healthcare systems. Unvaccinated health care workers remain at high risk of infection, and staff shortages continue to be a problem [3, 4].

Rapid, widely available and accurate COVID-19 laboratory tests allow effective case detection which is required for proper isolation of cases, contact tracing and quarantine [5]. Undocumented infections, which have been estimated to be as high as 90% of cases, have been described as the main driver of rapid dissemination of SARS-CoV-2 infection [6]. Although these undocumented cases are individually less likely to transmit virus, as a group they account for an estimated 79% of onward transmission events [7, 8]. Strategies including widespread, rapid, accurate and dynamically adaptive testing of infectious diseases are key for optimized pandemic response. Testing played a significant role in pandemic containment during the height of the pandemic, through phases of sporadic cases or clusters, even as overall transmission has declined, especially in areas with uneven geographic vaccine coverage [9]. Specifically, the establishment of asymptomatic testing programs to mitigate COVID-19 have been used in both healthcare and not healthcare settings. While SARS-CoV-2 has been described in fully vaccinated individuals [2, 10–13], the role and value of asymptomatic testing in this group is not well described [14]. In addition to widespread available testing, local and regional containment strategies are important to allow safe reopening of countries [15].

We describe a sustained and adaptable SARS-CoV-2 PCR-based screening program with rapid turnaround times, as part of a multilayered institutional COVID-19 mitigation strategy at a pediatric specialty hospital that treats immunocompromised patients and characterize the course of asymptomatic infections. We show the impact of the program in terms of number of asymptomatic infections detected and potential days of transmission, averted. This program spanned the initiation of a vaccination campaign allowing for the detection of SARS-CoV-2 in 16 asymptomatic, fully vaccinated employees and demonstrates a robust and flexible testing paradigm as a critical component of a multifaceted approach to disease control that can be deployed in future potential outbreaks.

## Methods

Nasal swab samples were placed in 3 mL universal transport media (UTM) and transported to the Clinical COVID Laboratory within 1–2 hours. Samples received by 3pm each day were immediately placed in lysis solution and processed by PCR for detection of SARS CoV-2 RNA. Testing was performed using one of three test systems: the NeuMoDx™ SARS-CoV-2 Assay, (Qiagen, Hilden, Germany), the Roche Cobas6800/8800 assay (Roche Diagnostics, (Risch-Rotkreuz, Switzerland), or the altona RealStar® SARS-COV-2 RT-PCR assay (altona Diagnostics, Hamburg, Germany), each of which had received emergency use authorization (EUA) by the US Food and Drug Administration (US FDA). All three methods had also undergone validation by the St. Jude Clinical COVID Laboratory and been shown to perform as expected, with comparable accuracy across all systems. Samples received after 3pm on a given day were stored at 4˚C 2–8˚C and processed the following morning. Most results were available within 12 hours of collection, and all were available within 24 hours. Remnant of samples positive for SARS-CoV-2

RNA were frozen at -70˚C and thawed at room temperature prior to sequencing. Paired-end sequencing was performed at least weekly on a MiSeq II (Illumina, Inc., San Diego, CA), using Swift Normalase® Amplicon Panel (SNAP) SARS-CoV-2 Additional Genome Coverage, and SARS-CoV-2 S Gene Panels (Swift Biosciences, Ann Arbor, MI) and analysis performed using both an internally developed computational pipeline (idCOV, Center for Applied Bioinformatics, St. Jude) and a commercial pipeline (COSMOSID, Rockville, MD). The sequencing process (wet and in silico portions) had undergone extensive validation using known SARS-CoV-2 variants, showing a high degree of accuracy. Variant determination in clinical samples was based on consensus results between the two analytic pipelines. Sequencing was performed every 30 days if serially positive samples were obtained from a given individual. Descriptive statistics, sum and median (range) for continuous variables, frequency (proportion) for categorical variables were summarized, along with 95% confidence interval whenever applied.

## Program description

St. Jude is located in downtown Memphis, Tennessee, a city that experienced a fluctuating number of SARS-CoV-2–positive individuals and COVID-19–related deaths throughout the study period. The local vaccination campaign also started over the study period with vaccine uptake uneven across the Memphis metropolitan area (main employee catchment area) [16]. The St. Jude multilayered COVID-19 mitigation strategy included the following actions: 1) Campus separation into clinical and research zones and implementation of strict access-control points to limit foot traffic in clinical buildings; 2) Delivery of repeated messaging to employees to not report to work if they had any COVID-19 symptoms or any contact with a SARS-CoV-2–positive individual; 3) Requirement that all employees wear medical masks and observe physical distancing when possible; 4) Performance of daily screening for COVID-19 symptoms and exposure for all patients, family members and persons on campus working in the clinical zone; 5) SARS-CoV-2 testing of anyone with symptoms or potential exposure, with specimen collection at an off-campus drive-through facility; 6) Weekly PCR-based testing of all asymptomatic employees working on campus; 7) Identification of work-related contacts of all employees testing positive and testing (within 24-hours) of all any employees believed to have experienced significant SARS-CoV-2 exposure; 8) COVID-19 vaccination following Tennessee state vaccination prioritization guidance starting December 18, 2020 [17]. Starting March 25, 2020, mid-turbinate nasal swab samples were collected from the on-campus workforce every 4–7 days and tested by PCR for SARS-CoV-2 RNA (up to 1576 tests per day). Sample collection was performed at a central, easily accessible location on campus. Through iterative process improvements led by the quality improvement team, the time to complete the check-in and nasal swab collection process was consistently 3 minutes or less. Within 12 weeks of starting the program, an institution-specific mobile app was developed and deployed for employees to check their symptoms and provide notification when the person was due for testing. The app was rapidly adopted by the workforce. Within 5 weeks of app launch, 90% of daily symptom checks were completed via the app, and this use has been sustained. Extensive education was provided to the entire workforce underscoring the importance of test-based screening to protect patients, families, and employees.

A dedicated COVID-19 testing laboratory was designed and constructed within the existing hospital CLIA-certified laboratory within a few weeks of the start of the pandemic. This space purposefully allowed for the use of both automated and non-automated nucleic acid amplification systems, facilitating a multi-assay approach that provided redundancy in the event of supply shortages or instrument malfunctions, while also providing surge capacity, in order to maintain rapid turn-around time in the face of transient, marked increases in testing volume.

Space design specifications allowed the use of appropriate personal protective equipment (PPE) and safe distancing of laboratory staff, while guarding against assay contamination events that could lead to false positive results. Positive results, reported within 2–24 hours, prompted case investigation, and contact tracing, which were completed within 6 hours. Follow up of all employees testing positive for SARS-CoV-2 was performed through weekly phone calls during the isolation period until employee met CDC criteria to return to work.

Routine SARS-CoV-2 PCR screening of asymptomatic employees was initially continued agnostic of COVID-19 vaccination status and SARS-CoV-2 PCR positive occurrences in vaccinated individuals were characterized and used to inform further screening and surveillance strategies that accounted for vaccination status.

The COVID-19 risk mitigation program assessment described herein was deemed exempt research by St Jude's institutional review board with a waiver of informed consent.

## Results

A total of 7497 self-reported asymptomatic workforce members were tested at least once for SARS-CoV-2 infection from March 18[th] 2020, to May 16[th], 2021 (>210,000 tests; range, 1–77 tests per individual). Overall, 50.55% (414 of 820) of the workforce members that tested positive for SARS-CoV-2 were diagnosed through the routine testing program, with an overall cumulative incidence of 5.64% ([95% CI 5.12–6.19%]) (Fig 1) and a weekly incidence ranging from 0–8.4 per 1000 persons. The data trends were similar to local county case occurrence patterns (Fig 1). Of 414 positive employees, 197 (47.58%) reported mild symptoms that they had initially dismissed or attributed to seasonal allergies until asked during case-investigation; 65 (15.70%) were pre-symptomatic and developed symptoms after testing [median 3 days (range 0–14 days)] from positive sample collection until symptom onset) and 152 (36.71%) remained asymptomatic. As positive individuals were instructed to immediately leave campus and begin quarantine, identification of these cases with routine SARS-CoV-2 testing averted 1800 days of potential on-campus transmission. This consisted of 280 days averted by identification of pre-symptomatic individuals who would otherwise have stayed on campus until symptom onset and 1520 days averted by identification of asymptomatic individuals (assuming 10 days of potential transmission per case). No suspected staff-to-patient or sustained staff-to-staff transmission of infection occurred during the duration of this test-based screening program.

Sixteen employees screened positive in the asymptomatic COVID-19 screening program 14 days or later after their second dose of mRNA COVID-19 vaccines among 5361 fully vaccinated employees (0.30%, [95% CI 0.17%-0.48%]); 13 were asymptomatic at time of testing and remained asymptomatic (81.25%) and three recalled having mild symptoms that they had attributed to allergies (18.75%) (Table 1). Four (25.0%) individuals had a history of domestic travel and 2 (12.5%) had a known COVID-19 exposure. While SARS-CoV-2 sequencing was attempted in all cases, it was inconclusive in 12 (75.0%), showed viruses of the B.1.1.7 lineage in three (18.75%), and B.1.526 in one worker (6.25%). In addition to 16 fully vaccinated employees who tested positive for SARS CoV-2 as part of asymptomatic workplace testing, during the same period 5 of 5361 fully vaccinated employees (0.09%, [95% CI 0.03%-0.22%]) tested positive for SARS-CoV-2 following development of symptoms consistent with COVID-19. The distribution of age or race was similar between those found SARS-CoV-2 PCR test positive by symptomatic screening versus those testing positive subsequent to symptoms. All five were female and received Pfizer. Only one (20.0%) of the latter workers had a history of travel. Interestingly, 4 out 5 (80.0%) had a known COVID-19 exposure. SARS-CoV-2 sequencing was available for two out of the 5 (40.0%) symptomatic workers and showed B.1.1.7-lineage viruses in both. None of these individuals was hospitalized due to COVID-19.

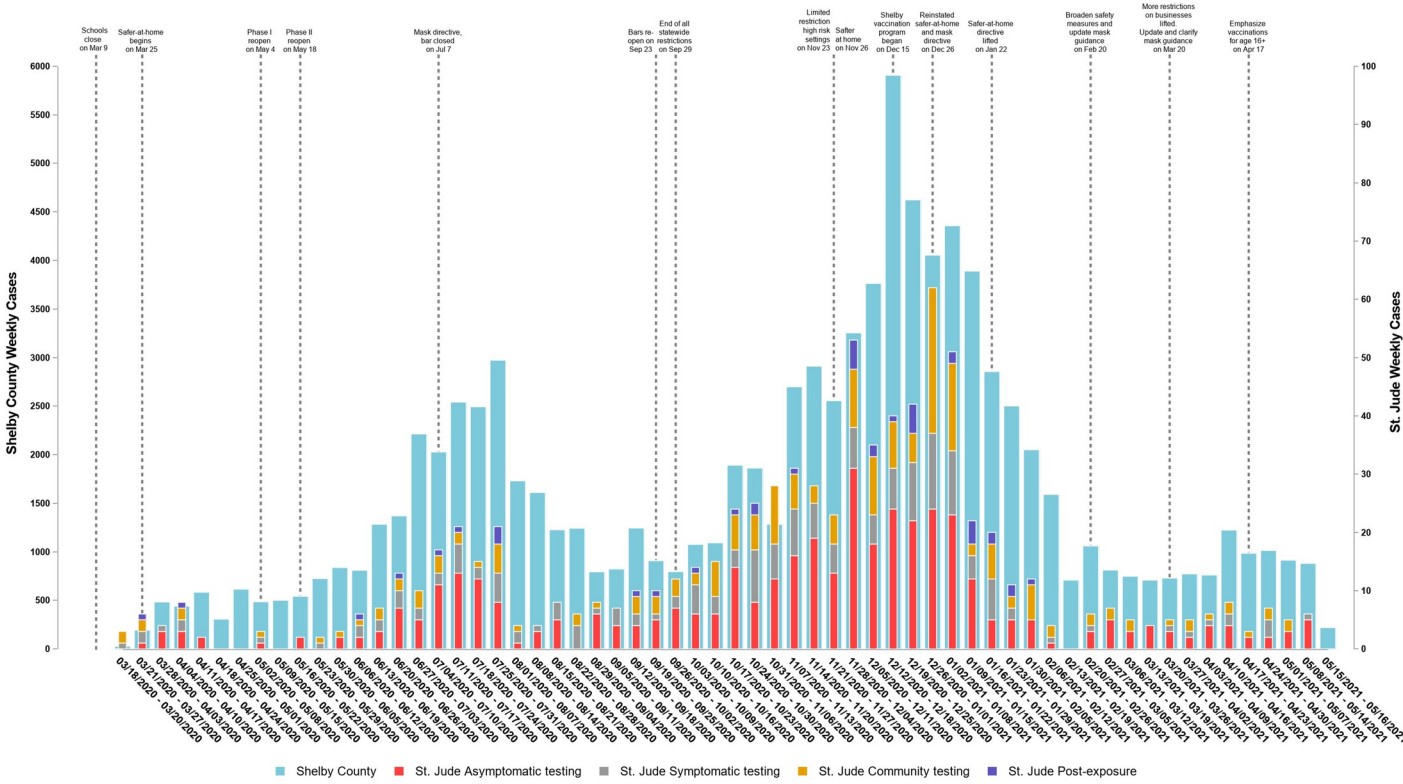

**Fig 1. Pulse of COVID-19 pandemic: Weekly number of confirmed COVID-19 cases in Shelby County and in individuals working at St. Jude Children's Research Hospital with a timeline of risk-mitigation directives and COVID-19 vaccine roll-out.** Cyan bar: weekly number of confirmed COVID-19 cases in Shelby County where St. Jude Children's Research Hospital is located. Data Source: Shelby County Health Department. Red bar: weekly number of confirmed COVID-19 employee cases at St. Jude Children's Research Hospital who were diagnosed by the institutional asymptomatic screening program. Gray bar: weekly number of confirmed COVID-19 employee cases at St. Jude who were diagnosed after presentation with symptoms to the institutional occupational health's symptom-based screening program. Orange bar: weekly number of confirmed COVID-19 employee cases at St. Jude who were diagnosed outside the institution and by community health care systems. Deep blue bar: weekly number of confirmed COVID-19 employee cases at St. Jude who were tested and diagnosed by St. Jude occupational health following reported COVID-19 exposure.

## Discussion

Asymptomatic infection has played a significant role in driving this pandemic. We show consistently over a 14-month period that half of COVID-19 infections were detected through an asymptomatic screening program, a third of whom never developed any symptoms during follow-up. Prompt detection and isolation of these cases substantially reduced the risk of potential workplace transmission. Mild symptomatology that was self-assessed by individuals as inconsequential or attributable to allergies was present in about half of cases detected in the asymptomatic screening program. The inaccurate self-assessment of symptoms could result from a combination of reasons including recall bias once a person was told they have tested positive for COVID-19, desensitization to symptom-based screening questions, or to baseline symptomatology related to non-COVID-19 related etiologies, such as environmental allergies. Regardless, it underscores the importance of asymptomatic test-based screening as part of a multilayered COVID-19 risk mitigation program. Asymptomatic screening in the workplace has been both recommended and described [4, 18]. However, sustainability and iterative adaption of such programs to the evolving course of the pandemic has not yet been described. Beyond symptomatic screening, physical distancing, and use of personal protective equipment, we show that routine, PCR-based, SARS-CoV-2 testing of asymptomatic employees is feasible,

**Table 1. Demographic characteristic and vaccination status of employees with COVID-19.**

| | Asymptomatic screening | | Symptomatic/ Post exposure testing[*] | |
| | N = 414 | | N = 406 | |
| | Unvaccinated | Vaccinated | Unvaccinated | Vaccinated |
| | N = 398 | N = 16 | N = 401 | N = 5 |
|---|---|---|---|---|
| Age (median, range) | 41.5 (18.75–72.26) | 55.74 (25.51–65.62) | 40.39 (18.06–71.53) | 43.33 (28.65 ~ 53.46) |
| Gender | | | | |
| Female | 242 (60.8%) | 8 (50%) | 283 (70.57%) | 4 (80%) |
| Male | 156 (39.2%) | 8 (50%) | 118 (29.43%) | 1 (20%) |
| Race | | | | |
| Black | 189 (47.73%) | 4 (25%) | 104 (26.53%) | 1 (20%) |
| White | 162 (40.91%) | 12 (75%) | 243 (61.99%) | 3 (60%) |
| Asian | 16 (4.04%) | 0 | 10 (2.55%) | 0 |
| Hispanic | 17 (4.29%) | 0 | 24 (6.12%) | 0 |
| Others | 12 (3.03%) | 0 | 11 (2.81%) | 1 (20%) |
| Days from second vaccine dose (median, range) | NA | 53 (14–94) | NA | 78 (63–92) |
| Type of Vaccine | | | | |
| Pfizer-BioNTech | NA | 15 (93.75%) | NA | 5 (100%) |
| Moderna | NA | 1 (6.25%) | NA | 0 |

[*]Includes testing due to onset of symptoms consistent with COVID-19 or after exposure to someone with COVID-19.

sustainable, and minimizes the risk of workplace transmission. Such screening monitors the local pulse of the COVID-19 pandemic and can provide valuable data to assess the ongoing impact of COVID-19 vaccination and SARS-CoV-2 variants, inform risk mitigation, and guide adaptive operational planning including titration of screening strategies over time based on infection risk modifiers such as vaccination.

A significant decrease in both symptomatic and asymptomatic SARS-CoV-2 infections was seen following the initiation of COVID-19 vaccinations among our workforce (a majority receiving BNT162b2 vaccine [Pfizer-BioNTech]) [17]. We describe from our experience 21 cases (3.9 per 1000 fully vaccinated individuals) of SARS-CoV-2 test positivity 14 days or more after their second dose of mRNA vaccines and who fully recovered without complications. While most of our cases were asymptomatic, the CDC reported only 27% of 10,262 SARS-CoV-2 of vaccine breakthrough infections to be asymptomatic [11]. The discrepancy with our results, likely is due to lack of testing of asymptomatic individuals outside of screening programs and an undercount from the CDC given that reporting is both passive and voluntary. Detection of variants of concern was reflective of variants being seen at that time in the community. While all were advised to follow isolation guidance, the clinical relevance of these, primarily asymptomatic post-vaccination cases remain unclear. Based on the relatively low case incidence, low incidence of clinical symptoms, and an inability to sequence the virus (potentially indicating a low viral burden), the cost-risk-benefit did not favor continued testing in asymptomatic employees who had completed COVID-19 vaccination and the practice was thus discontinued. We suggest that SARS-CoV-2 PCR positive tests in individuals who remain asymptomatic should not be grouped under the common term of "breakthrough infections" since the clinical implications both for the host and from a transmission standpoint are likely different than in those who have not been fully vaccinated. We continue to monitor the evolution of SARS-CoV-2 variants and will resume SARS-CoV-2 PCR testing in those vaccinated if clinically relevant infections are noted or transmission of infection from those vaccinated is

described. Routine testing of unvaccinated members of our workforce remains important and continues.

Over the past 14 months, most COVID-19 infections in our workforce have been non-work-related and, as shown in Fig 1, have followed community incidence patterns, the latter, in turn, correlating with public health directives and interventions. With this sustained, adaptive, SARS-CoV-2 PCR testing-enhanced COVID-19 screening program we have demonstrated the ability to potentially modify institutional transmission risk. That risk is particularly pertinent for the majority immunocompromised population we treat. This program also provided the ability to iteratively scale back testing during vaccination roll-out. Frequent SARS-CoV-2 PCR testing of asymptomatic individuals should be prioritized to guide institutional risk-mitigation efforts, inform ongoing surveillance and prompt interventions.

Strong preparedness systems coupled with decisive responses are key for a successful pandemic response [5]. As such, accessibility, convenience, leadership commitment and communication contributed to the success and sustainability of our program [17]. While the program we implemented was designed specifically for an institutional COVID-19 pandemic response, many of the systems put in place and lessons learned are generalizable [5]. The ability to remove potentially infectious individuals promptly from a workplace would take on even greater importance during an outbreak of a more virulent pathogen. Although the specific tests utilized would be different, the systems developed to screen individuals, maintain sample integrity, and provide a barrier between infectious and susceptible individuals would be interchangeable [5]. The primary testing systems used for PCR were open platforms, enabling their rapid redeployment to detect other pathogens. By nature, infectious diseases including SARS CoV-2 infection show variable disease severity in a population, often ranging from asymptomatic to fatal and their optimal control is unlikely through symptom-based testing alone. A robust and flexible testing paradigm will remain a critical component of a multifaceted approach to disease control in future potential outbreaks.

## Conclusions

Routine SARS CoV-2 PCR based screening enhanced, multicomponent COVID-19 risk mitigation program is feasible, sustainable and can be adapted through the phases of the pandemic. Such a program not only decreases risk of SARS CoV-2 transmission but can aid institutions in deciding to expand or contract operations throughout the pandemic and in turn help direct and sustain recovery efforts as vaccination becomes more accessible and acceptable.

## Acknowledgments

The authors would like to acknowledge: The Occupational Health team, the Infection Prevention & Control team, and the COVID-19 Case Investigation and Contact Tracing team, for their role in triage of employees with symptoms, case investigation, contact tracing, and follow-up. We thank staff from the St Jude Departments of Infectious Diseases, Immunology, Therapeutics Production and Quality, and the Children's GMP, LLC. who made crucial contributions to enable testing to start in March 2020. We also thank St. Jude's nurses who established, standardized, and operated the sample-collection process. We thank the St. Jude Molecular Microbiology and COVID Laboratories and the Pathology Informatics Team for assay design, validation, implementation, testing and reporting.

The St. Jude COVID-19 Employee Screening Program Team: Lisa Kercher, Ph.D.,[1] Thomas P. Fabrizio, Ph.D.,[1] John Franks,[1] Pamela Freiden,[1] Bridgett Sharp,[1] Robert C. Mettelman Ph. D.,[2] Austin Springer Ph.D.,[3] Motomi Mori Ph.D., M.B.A.,[4] Sandra R. Dennis,[3] Yilun Sun,[4] Yin Su,[4] Vijaya Rajagopal,[5] Jason Massey,[5] Paul E. Mead, Ph.D.,[5] Maria P. Gann,[5] Zhengming Gu,

Ph.D.,[5] Jessica Brazelton de Cardenas, Ph.D.,[5] Bryan Mathieson,[6] RN, Kari Lahmon, RN,[6] and Sri Suganda.[5]

[1] Departments of Infectious Diseases, St. Jude Children's Research Hospital, Memphis, Tennessee, United States of America

[2] Departments of Immunology, St. Jude Children's Research Hospital, Memphis, Tennessee, United States of America

[3] Departments of Human Resources, St. Jude Children's Research Hospital, Memphis, Tennessee, United States of America

[4] Departments of Biostatistics, St. Jude Children's Research Hospital, Memphis, Tennessee, United States of America

[5] Departments of Pathology, St. Jude Children's Research Hospital, Memphis, Tennessee, United States of America

[6] Departments of Nursing Administration, St. Jude Children's Research Hospital, Memphis, Tennessee, United States of America

Lead author: Kari Lahmon, RN; email: kari.lahmon@stjude.org

## Author Contributions

**Conceptualization:** Diego R. Hijano, James M. Hoffman, Hana Hakim, Richard J. Webby, Randall T. Hayden, Aditya H. Gaur.

**Data curation:** Diego R. Hijano, Li Tang.

**Formal analysis:** Li Tang.

**Investigation:** Stacey L. Schultz-Cherry, Paul G. Thomas, Richard J. Webby.

**Methodology:** James M. Hoffman, Randall T. Hayden, Aditya H. Gaur.

**Project administration:** Randall T. Hayden, Aditya H. Gaur.

**Supervision:** James M. Hoffman, Randall T. Hayden, Aditya H. Gaur.

**Writing – original draft:** Diego R. Hijano.

**Writing – review & editing:** James M. Hoffman, Li Tang, Stacey L. Schultz-Cherry, Paul G. Thomas, Hana Hakim, Richard J. Webby, Randall T. Hayden, Aditya H. Gaur.

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
