## [Decision Letter · Decision Letter 0]

15 Mar 2022

PONE-D-21-19497An Adaptive, Asymptomatic SARS-CoV-2 Workforce Screening Program Providing Real-Time, Actionable Monitoring of the COVID-19 Pandemic.PLOS ONE

Dear Dr. Hijano,

Thank you for submitting your manuscript to PLOS ONE. After careful consideration, we feel that it has merit but does not fully meet PLOS ONE’s publication criteria as it currently stands. Therefore, we invite you to submit a revised version of the manuscript that addresses the points raised during the review process.

We look forward to receiving your revised manuscript.

Kind regards,

Dong Keon Yon, MD, FACAAI

Academic Editor

PLOS ONE

2. Please amend your current ethics statement to include the full name of the ethics committee/institutional review board(s) that waived your specific study.

Furthermore, please provide additional information regarding whether informed consent was taken from the participants or whether the IRB specifically waived the need for informed consent.

“SSC, PGT, and RJW are partially supported by the National Institutes of Health, under CEIRS contract no. HHSN272201400006C, SSC under CIVIC contract no.75N93019C00052 and PGT under R01 AI121832. The described work was partially supported by the American Lebanese Syrian Associated Charities (ALSAC).”

Please state what role the funders took in the study.  If the funders had no role, please state: ""The funders had no role in study design, data collection and analysis, decision to publish, or preparation of the manuscript.

“SSC, PGT, and RJW are partially supported by the National Institutes of Health, under CEIRS contract no. HHSN272201400006C, SSC under CIVIC contract no.75N93019C00052 and PGT under R01 AI121832. The described work was partially supported by the American Lebanese Syrian Associated Charities (ALSAC).”

5. Thank you for stating the following in the Funding Section of your manuscript:

“SSC, PGT, and RJW are partially supported by the National Institutes of Health, under CEIRS contract no. HHSN272201400006C, SSC under CIVIC contract no.75N93019C00052 and PGT under R01 AI121832. The described work was partially supported by the American Lebanese Syrian Associated Charities (ALSAC).”

“SSC, PGT, and RJW are partially supported by the National Institutes of Health, under CEIRS contract no. HHSN272201400006C, SSC under CIVIC contract no.75N93019C00052 and PGT under R01 AI121832. The described work was partially supported by the American Lebanese Syrian Associated Charities (ALSAC).”

6. Thank you for stating the following in the Competing Interests section:

“PGT is on the Scientific Advisory Board for Cytoagents Inc. RTH has served on advisory boards for Roche Molecular and Quidel Corporation. All other authors have no conflicts of interest.”

7. One of the noted authors is a group or consortium [L Kercher TP Fabrizio J Franks P Freiden B Sharp, RC Mettelman A Springer M Mori SR Dennis Y Sun Y Su V Rajagopal J Massey PE Mead MP Gann Z Gu J Brazelton de Cardenas B Mathieson K Lahmon S Suganda]. In addition to naming the author group, please list the individual authors and affiliations within this group in the acknowledgments section of your manuscript. Please also indicate clearly a lead author for this group along with a contact email address.

8. Please include a copy of Table 1 which you refer to in your text on page 6.

Additional Editor Comments:

I am sorry for the delay response of your paper due to the absence of an your assigning editor. Do the revision quickly please.

Reviewers' comments:

Reviewer's Responses to Questions

**Comments to the Author**

1. Is the manuscript technically sound, and do the data support the conclusions?

Reviewer #1: Partly

Reviewer #2: Yes

2. Has the statistical analysis been performed appropriately and rigorously? 

Reviewer #1: No

Reviewer #2: Yes

3. Have the authors made all data underlying the findings in their manuscript fully available?

Reviewer #1: Yes

Reviewer #2: Yes

4. Is the manuscript presented in an intelligible fashion and written in standard English?

Reviewer #1: Yes

Reviewer #2: Yes

5. Review Comments to the Author

Reviewer #1: An Adaptive, Asymptomatic SARS-CoV-2 Workforce Screening Program

Editorial comments

• Lines 65-66: Local and regional containment strategies are important to allow safe reopening of the country [10]. Which country? Or do you mean countries?

• Line 66: …. …regional containment strategies are important to allow safe reopening of the country [10]. Line 66: SARCoV-2 has been described in fully vaccinated individuals. There is a jump from one idea to the next. There is a need for connector between these two sentences.

• Line 70: SARS-CoV-2 PCR-based.

• Line 71: turnaround times aimed at asymptomatic

• Line 154: ….CoV 2 PCR testp positive….

• Line 164: “One in two”, may be half?

General comments

• According to Line 7, the aim of this paper is to “describe a sustained and adaptable SARS-CoV-2 PCR based testing program with rapid turnaround aimed at asymptomatic workforce members, as part of a multilayered institutional COVID-19 mitigation strategy at a pediatric specialty hospital that treats immunocompromised patients”. This is not a sufficient aim for a paper. For the paper to be impactful, it should not only describe but also critically assess or analyze the program.

• The methods section, as currently presented, does not describe the methods used in the paper or study. Rather, the section describes the Asymptomatic SARS-CoV-2 Workforce Screening Program as part of a wider Covid-19 management program. Therefore, the section should correctly be given the heading Program description. On the contrary, the results section presents various statistical measures as part of the evaluation of the program. The methods used in the calculation of the statistical parameters and the overall analysis or evaluation of the program should be presented in the methods section. How were asymptomatic workers identified, how were vaccinated employees identified? How was sequencing conducted? How was follow-up conducted?

• The paper has some time-to-event data in form of “median time from second vaccine dose to positive SARS CoV-2 PCR test. How was this assessed? Was a survival analysis conducted

• Why are demographic data only provided for vaccinated employees?

• Is it possible to tabulate some of the information provided in the results section?

• In the results section, vaccination results are presented by name or type of vaccine. Is it not necessary to discuss the results by type of vaccine.

Reviewer #2: This is excellent paper. I read it with great interest and I have no further comments.

#1. Update your paper's reference on the latest date.

#2. Please discuss about the vaccination program (booster or and so on) breifly.

6. PLOS authors have the option to publish the peer review history of their article (what does this mean?). If published, this will include your full peer review and any attached files.

Reviewer #1: **Yes: **Wells Utembe

Reviewer #2: No

---

## [Author Response · Author response to Decision Letter 0]

11 Apr 2022

April 9, 2022

Re: Response to reviewer comments received March 21st , 2022, for manuscript PONE-D-21-19497: “An Adaptive, Asymptomatic SARS-CoV-2 Workforce Screening Program Providing Real-Time, Actionable Monitoring of the COVID-19 Pandemic.”

We thank the reviewers and editors for their time and insightful comments (noted in italics). Please find below a point-by-point response.

Editorial comments

Lines 65-66: Local and regional containment strategies are important to allow safe reopening of the country [10]. Which country? Or do you mean countries? 

Response: We meant countries. This has been corrected.

Line 66: …. …regional containment strategies are important to allow safe reopening of the country [10]. Line 66: SARCoV-2 has been described in fully vaccinated individuals. There is a jump from one idea to the next. There is a need for connector between these two sentences. 

Response: This section has been edited as follows: 

“Testing played a significant role in pandemic containment during the height of the pandemic, through phases of sporadic cases or clusters, even as overall transmission has declined, especially in areas with uneven geographic vaccine coverage [9]. Specifically, the establishment of asymptomatic testing programs to mitigate COVID-19 have been used in both healthcare and not healthcare settings. While SARS-CoV-2 has been described in fully vaccinated individuals [2, 10-13], the role and value of asymptomatic testing in this group is not well described, although, the US CDC has recommended halting this workplace practice [14]. In addition to widespread available testing, local and regional containment strategies are important to allow safe reopening of countries [15].” 

Line 70: SARS-CoV-2 PCR-based.

Response: This has been corrected.

Line 71: turnaround times aimed at asymptomatic

Response: This has been corrected.

Line 154: ….CoV 2 PCR testp positive….

Response: This has been corrected.

Line 164: “One in two”, may be half? 

Response: This has been modified.

General comments

According to Line 7, the aim of this paper is to “describe a sustained and adaptable SARS-CoV-2 PCR based testing program with rapid turnaround aimed at asymptomatic workforce members, as part of a multilayered institutional COVID-19 mitigation strategy at a pediatric specialty hospital that treats immunocompromised patients”. This is not a sufficient aim for a paper. For the paper to be impactful, it should not only describe but also critically assess or analyze the program.

Response: We have more than described the program. We have shown the feasibility of an asymptomatic SARS-CoV-2 workforce screening program, its impact on detection of asymptomatic infections and resultant potential days of transmission averted underscoring its values as an important component of a multifaceted risk mitigation strategy and its iterative adaption based on the pulse of the pandemic. The last paragraph of the introduction now reads:

“We describe a sustained and adaptable SARS-CoV-2 PCR-based screening program with rapid turnaround times, as part of a multilayered institutional COVID-19 mitigation strategy at a pediatric specialty hospital that treats immunocompromised patients and characterize the course of asymptomatic infections. We show the impact of the program in terms of number of asymptomatic infections detected and potential days of transmission, averted. This program spanned the initiation of a vaccination campaign allowing for the detection of SARS-CoV-2 in 16 asymptomatic, fully vaccinated employees and demonstrates a robust and flexible testing paradigm as a critical component of a multifaceted approach to disease control that can be deployed in future potential outbreaks.”

The methods section, as currently presented, does not describe the methods used in the paper or study. Rather, the section describes the Asymptomatic SARS-CoV-2 Workforce Screening Program as part of a wider Covid-19 management program. Therefore, the section should correctly be given the heading Program description. On the contrary, the results section presents various statistical measures as part of the evaluation of the program. The methods used in the calculation of the statistical parameters and the overall analysis or evaluation of the program should be presented in the methods section. 

Response: A methods section has been included with this information and the content describing the program was moved to a section titled “Program description” as suggested by the reviewer.

How were asymptomatic workers identified, how were vaccinated employees identified? 

Response: Initially, we had trained screeners asking for symptoms upon employees entering the restricted access points to campus. Eventually, an app was developed, and most employees adopted the use of the app to self-screen for symptoms daily. In addition, Occupational Health would ask presence or absence of symptoms in those who tested positive for SARS-CoV-2.

How was sequencing conducted? 

Response: A methods section has been included with information related to SARS-CoV-2 PCR testing and sequencing.

How was follow-up conducted?

Response: The following sentence was added in the Program description: 

“Follow up of all employees testing positive for SARS-CoV-2 was performed through weekly phone calls during the isolation period until employee met CDC criteria to return to work.”

The paper has some time-to-event data in form of “median time from second vaccine dose to positive SARS CoV-2 PCR test. How was this assessed? Was a survival analysis conducted? 

Response: A formal survival analysis was not done given the relatively small number of events. Data shown is using descriptive statistics. 

Why are demographic data only provided for vaccinated employees?

Response: The following table has been added including information on unvaccinated employees.

 Asymptomatic screening

N=414 Symptomatic/ Post exposure testing*

N=406

 Unvaccinated

N=398 Vaccinated

N=16 Unvaccinated

N=401 Vaccinated

N=5

Age (median, range) 41.5 (18.75-72.26) 55.74 (25.51- 65.62) 40.39 (18.06-71.53) 43.33 (28.65 ~ 53.46)

Gender

Female 

Male 

 242 (60.8%) 8 (50%) 283 (70.57%) 4 (80%)

 156 (39.2%) 8 (50%) 118 (29.43%) 1 (20%)

Race

Black

White

Asian

Hispanic

Others 

 189 (47.73%) 4 (25%) 104 (26.53%) 1 (20%)

 162 (40.91%) 12 (75%) 243 (61.99%) 3 (60%)

 16 (4.04%) 0 10 (2.55%) 0

 17 (4.29%) 0 24 (6.12%) 0

 12 (3.03%) 0 11 (2.81%) 1 (20%)

Days from second vaccine dose (median, range) NA 53 (14-94) NA 78 (63-92)

Type of Vaccine

Pfizer-BioNTech

Moderna 

 NA 15 (93.75%) NA 5 (100%)

 NA 1 (6.25%) NA 0

*Includes testing due to onset of symptoms consistent with COVID-19 or after exposure to someone with COVID-19

Is it possible to tabulate some of the information provided in the results section?

Response: A table has been added with the requested demographic information.

In the results section, vaccination results are presented by name or type of vaccine. Is it not necessary to discuss the results by type of vaccine? 

Response: We appreciate reviewer’s comment. All SARS-CoV-2 cases in fully vaccinated employees described in the manuscript were among employees who received an mRNA vaccine, with only one receiving Moderna. The main reason is that Pfizer-BioNTech was the vaccine that was widely available to our institution and does not reflect any differences related to vaccine effectiveness.

---

## [Decision Letter · Decision Letter 1]

26 Apr 2022

An Adaptive, Asymptomatic SARS-CoV-2 Workforce Screening Program Providing Real-Time, Actionable Monitoring of the COVID-19 Pandemic.

PONE-D-21-19497R1

Dear Dr. Hijano,

We’re pleased to inform you that your manuscript has been judged scientifically suitable for publication and will be formally accepted for publication once it meets all outstanding technical requirements.

Kind regards,

Dong Keon Yon, MD, FACAAI

Academic Editor

PLOS ONE

Additional Editor Comments (optional):

Congratulation on your mesmerizing paper.

Reviewers' comments:

Reviewer's Responses to Questions

**Comments to the Author**

1. If the authors have adequately addressed your comments raised in a previous round of review and you feel that this manuscript is now acceptable for publication, you may indicate that here to bypass the “Comments to the Author” section, enter your conflict of interest statement in the “Confidential to Editor” section, and submit your "Accept" recommendation.

Reviewer #1: All comments have been addressed

2. Is the manuscript technically sound, and do the data support the conclusions?

Reviewer #1: Yes

3. Has the statistical analysis been performed appropriately and rigorously? 

Reviewer #1: Yes

4. Have the authors made all data underlying the findings in their manuscript fully available?

Reviewer #1: Yes

5. Is the manuscript presented in an intelligible fashion and written in standard English?

Reviewer #1: Yes

6. Review Comments to the Author

Reviewer #1: The paper can now be accepted as the authors have restructured the paper and addressed all my concerns.

7. PLOS authors have the option to publish the peer review history of their article (what does this mean?). If published, this will include your full peer review and any attached files.

Reviewer #1: No

---

## [Editor Report · Acceptance letter]

28 Apr 2022

PONE-D-21-19497R1 

An adaptive, asymptomatic SARS-CoV-2 workforce screening program providing real-time, actionable monitoring of the COVID-19 pandemic. 

Dear Dr. Hijano:

I'm pleased to inform you that your manuscript has been deemed suitable for publication in PLOS ONE. Congratulations! Your manuscript is now with our production department. 

Kind regards, 

on behalf of

Dr. Dong Keon Yon 

Academic Editor

PLOS ONE